# FL-WBC: Enhancing Robustness against Model Poisoning Attacks in Federated Learning from a Client Perspective

**Jingwei Sun[1], Ang Li[1], Louis DiValentin[2], Amin Hassanzadeh[2],**
**Yiran Chen[1], Hai Li[1]**
[1]Department of Electrical and Computer Engineering, Duke University
[2]Security R&D, Accenture Labs, Accenture
[1]{jingwei.sun, ang.li630, yiran.chen, hai.li}@duke.edu,
[2]{louis.divalentin, amin.hassanzadeh}@accenture.com

## Abstract

Federated learning (FL) is a popular distributed learning framework that trains a global model through iterative communications between a central server and edge devices. Recent works have demonstrated that FL is vulnerable to model poisoning attacks. Several server-based defense approaches (e.g. robust aggregation) have been proposed to mitigate such attacks. However, we empirically show that under extremely strong attacks, these defensive methods fail to guarantee the robustness of FL. More importantly, we observe that as long as the global model is polluted, the impact of attacks on the global model will remain in subsequent rounds even if there are no subsequent attacks. In this work, we propose a client-based defense, named *White Blood Cell for Federated Learning (FL-WBC)*, which can mitigate model poisoning attacks that have already polluted the global model. The key idea of FL-WBC is to identify the parameter space where long-lasting attack effect on parameters resides and perturb that space during local training. Furthermore, we derive a certified robustness guarantee against model poisoning attacks and a convergence guarantee to FedAvg after applying our FL-WBC. We conduct experiments on FasionMNIST and CIFAR10 to evaluate the defense against state-of-the-art model poisoning attacks. The results demonstrate that our method can effectively mitigate model poisoning attack impact on the global model within 5 communication rounds with nearly no accuracy drop under both IID and non-IID settings. Our defense is also complementary to existing server-based robust aggregation approaches and can further improve the robustness of FL under extremely strong attacks. Our code can be found at https://github.com/jeremy313/FL-WBC.

## 1 Introduction

Federated learning (FL) [1, 2] is a popular distributed learning approach that enables a number of edge devices to train a shared model in a federated fashion without transferring their local training data. However, recent works [3–12] show that it is easy for edge devices to conduct model poisoning attacks by manipulating local training process to pollute the global model through aggregation.

Depending on the adversarial goals, model poisoning attacks can be classified as *untargeted model poisoning attacks* [3–6], which aim to make the global model indiscriminately have a high error rate on any test input, or *targeted model poisoning attacks* [7–12], where the goal is to make the global model generate attacker-desired misclassifications for some particular test samples. Our work focuses

35th Conference on Neural Information Processing Systems (NeurIPS 2021).

on the *targeted model poisoning attacks* introduced in [11, 12]. In this attack, malicious devices share a set of data points with dirty labels, and the adversarial goal is to make the global model output the same dirty labels given this set of data as inputs. Our work can be easily extended to many other model poisoning attacks (e.g., backdoor attacks), which shall be discussed in §4.

Several studies have been done to improve the robustness of FL against model poisoning attacks through robust aggregations [13–17], clipping local updates [7] and leveraging the noisy perturbation [7]. These defensive methods focus on only preventing the global model from being polluted by model poisoning attacks during the aggregation. However, we empirically show that these server-based defenses fail to guarantee the robustness when attacks are extremely strong. More importantly, we observe that as long as the global model is polluted, the impact of attacks on the global model will remain in subsequent rounds even if there are no subsequent attacks, and can not be mitigated by these server-based defenses. Therefore, an additional defense is needed to mitigate the poisoning attacks that cannot be eliminated by robust aggregation and will pollute the global model, which is the goal of this paper.

To achieve this goal, we first propose a quantitative estimator named *Attack Effect on Parameter (AEP)*. It estimates the effect of model poisoning attacks on global model parameters and infers information about the susceptibility of different instantiations of FL to model poisoning attacks. With our quantitative estimator, we explicitly show the long-lasting attack effect on the global model. Based on our analysis, we design a client-based defense named *White Blood Cell for Federated Learning (FL-WBC)*, as shown in Figure 1, which can mitigate the model poisoning attacks that have already polluted the global model. FL-WBC differs from previous server-based defenses in mitigating the model poisoning attack that has already broken through the server-based defenses and polluted the global model. Thus,

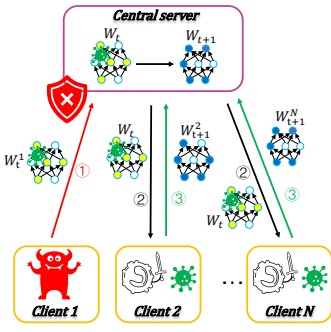

Figure 1: Overview of FL-WBC.

our client-based defense is complementary to current server-based defense and enhances the robustness of FL against the model poisoning attack, especially against the extremely strong attacks that can not be mitigated during the aggregation. We evaluate our defense on Fashion-MNIST [18] and CIFAR10 [19] against the model poisoning attack [11] under IID (identically independently distributed) and non-IID settings. The results demonstrate that FL-WBC can effectively mitigate the attack effect on the global model in 1 communication round with nearly no accuracy drop under IID settings, and within 5 communication rounds for non-IID settings, respectively. We also conduct experiments by integrating the robust aggregation with FL-WBC. The results show that even though the robust aggregation is ineffective under extremely strong attacks, the attack can still be efficiently mitigated by applying FL-WBC.

Our key contributions are summarized as follows:

- To the best of our knowledge, this is the first work to quantitatively assess the effect of model poisoning attack on the global model in FL. Based on our proposed estimator, we reveal the reason for the long-lasting effect of a model poisoning attack on the global model.

- We design a defense, which is also the first defense to the best of our knowledge, to effectively mitigate a model poisoning attack that has already polluted the global model. We also derive a robustness guarantee in terms of $AEP$ and a convergence guarantee to FedAvg when applying our defense.

- We evaluate our defense on Fashion-MNIST and CIFAR10 against state-of-the-art model poisoning attacks. The results show that our proposed defense can enhance the robustness of FL in an effective and efficient way, i.e., our defense defends against the attack in fewer communication rounds with less model utility degradation.

## 2   Related work

**Model poisoning attacks in FL**   Model poisoning attack can be *untargeted* [3–6] or *targeted* [7–12]. *Untargeted model poisoning attacks* aim to minimize the accuracy of the global model indiscriminately for any test input. For *targeted model poisoning attacks*, the malicious goal is to make the global model misclassify the particular test examples as the attacker-desired target class

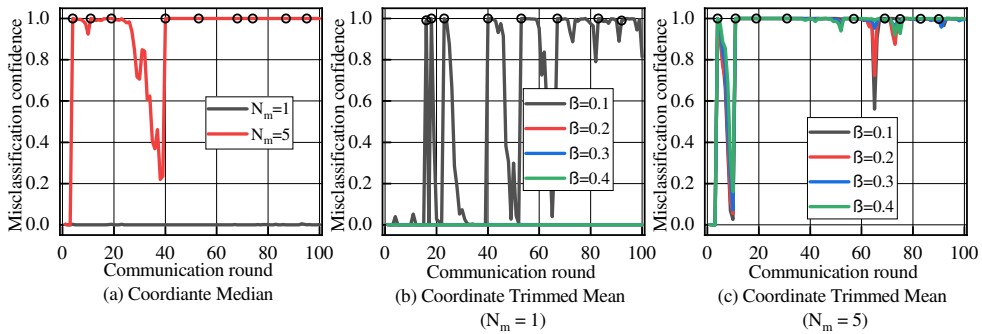

Figure 2: Defense performance of Coordinate Median aggregation and Coordinate Trimmed Mean aggregation. The black circle denotes the adversarial rounds for all the strategies in the figure.

in its prediction. An adversary using this approach can implant hidden backdoors into the global model so that the images with a trojan trigger will be classified as attacker-desired labels, known as a backdoor attack [7–10]. Another type of targeted model poisoning attack is introduced in [11, 12], which aims to fool the global model to produce adversarial misclassification on a set of chosen inputs with high confidence. Our work focuses on the targeted model poisoning attacks in [11, 12].

**Mitigate model poisoning attacks in FL**    A number of robust aggregation approaches have been proposed to mitigate data poisoning attacks while retaining the performance of FL. One typical approach is to detect and down-weight the malicious client's updates on the central server side [13–16], thus the attack effects on training performance can be diminished. The central server calculates coordinate-wise median or coordinate-wise trimmed mean for local model updates before performing aggregation [13]. Similarly, [14] suggests applying geometric median to local updates that are uploaded to the server. Meanwhile, some heuristic-based aggregation rules [20, 21, 3, 22, 23] have been proposed to cluster participating clients into a benign group and a malicious group, and then perform aggregation on the benign group only. FoolsGold [20] assumes that benign clients can be distinguished from attackers by observing the similarity between malicious clients' gradient updates, but Krum [21, 3] utilizes the similarity of benign clients' local updates instead. In addition, [7, 24] show that applying differential privacy to the aggregated global model can improve the robustness against model poisoning attacks. All these defensive methods are deployed at the server side and their goals are to mitigate model poisoning attacks during aggregation. Unfortunately, often in extreme cases (e.g. attackers occupy a large proportion of total clients), existing robust aggregation methods fail to prevent the aggregation from being polluted by the malicious local updates showing that it is not sufficient to offer defense via aggregation solely. Thus, there is an urgent necessity to design a novel local training method in FL to enhance its robustness against model poisoning attacks at the client side, which is complementary to existing robust aggregation approaches.

## 3   Motivation

Although current server-based defense approaches can defend against model poisoning attacks under most regular settings, it is not clear whether their robustness can still be guaranteed under extremely strong attacks, i.e., with significantly larger numbers of malicious devices involved in training. To investigate the robustness of current methods under such challenging but practical settings, we evaluate Coordinate Median aggregation (CMA) and Coordinate Trimmed Mean aggregation (CTMA) [13] on the model poisoning attack with Fashion-MNIST dataset, which is performed by following the settings in [11]. The goal of the attacks is to make the global model misclassify some specified data samples as target classes. In this experiment, we denote a communication round as an adversarial round $t_{adv}$ when malicious devices participate in the training, and $N_m$ malicious devices would participate in training at adversarial rounds. We assume that there are 10 devices involved in training in each round, but increase $N_m$ from 1 to 5 to vary the strength of the attacks. We conduct experiments under IID setting and the training data is uniformly distributed to 100 devices. The model architecture can be found in Table 3. For training, we set local epoch $E$ as 1 and batch size $B$ as 32. We apply SGD optimizer and set the learning rate $\eta$ to 0.01. The results of confidence that the global model would miss-classify the poisoning data point are shown in Figure  2.

The results show that the effectiveness of both CMA and CTMA dramatically degrades when there are 50% of malicious devices in the adversarial rounds. It is worthy noting that the attack impact on model performance will remain for subsequent rounds even if no additional attacks occur. We observe the same phenomenon in alternative robust aggregation approaches, and more detailed results are presented in §7. Therefore, in order to build a more robust FL system, it is necessary to instantly mitigate the impact of model poisoning attack as long as the global model is polluted by malicious devices. This has motivated us to design FL-WBC to ensure sufficient robustness of FL even under extremely strong attacks.

## 4  Model Poisoning Attack in FL

To better understand the impact of model poisoning attacks in FL scenarios, we first need to theoretically analyze how the poisoning attack affects the learning process and provide a mathematical estimation to quantitatively assess the attack effect on model parameters. During this process we come to a deeper understanding of the reasons for the persistence of the attack effect observed in §3. Without loss of generality, we employ FedAvg [1], the most widely applied FL algorithm as the representative FL method throughout this paper.

### 4.1  Problem Formulation

The learning objective of FedAvg is defined as:

$$W = \min_{W} \{F(W) \triangleq \sum_{k=1}^{N} p^k F^k(W)\}, \tag{1}$$

where $W$ is the weights of the global model, $N$ represents the number of devices, $F^k$ is the local objective of the $k$-th device, $p^k$ is the weight of the $k$-th device, $p^k \geq 0$ and $\sum_{k=1}^{N} p^k = 1$.

Equation 1 is solved in an iterative device-server communication fashion. For a given communication round (e.g. the $t$-th), the central server first randomly selects $K$ devices to compose a set of participating devices $\mathbb{S}_t$ and then broadcasts the latest global model $W_{t-1}$ to these devices. Afterwards, each device (e.g. the $k$-th) in $\mathbb{S}_t$ performs $I$ iterations of local training using their local data. However, the benign devices and malicious devices perform the local training in different manners. Specifically, if the $k$-th device is benign, in each iteration (e.g. the $i$-th), the local model $W_{t,i}^k$ on the $k$-th device is updated following:

$$W_{t,i+1}^k \leftarrow W_{t,i}^k - \eta_{t,i} \nabla F^k(W_{t,i}^k, \xi_{t,i}^k), \tag{2}$$

where $\eta_{t,i}$ is the learning rate, $\xi_{t,i}^k$ is a batch of data samples uniformly chosen from the $k$-th device and $W_{t,0}^k$ is initialized as $W_{t-1}$. In contrast, if the $k$-th device is malicious, the local model $W_{t,i}^k$ is updated according to:

$$W_{t,i+1}^k \leftarrow W_{t,i}^k - \eta_{t,i}[\alpha \nabla F^k(W_{t,i}^k, \xi_{t,i}^k) + (1-\alpha)\nabla F_M(W_{t,i}^k, \pi_{t,i})], \tag{3}$$

where $F_M$ is the malicious objective shared by all the malicious devices. $D_M$ is a malicious dataset that consists of the data samples following the same distribution as the benign training data but with adversarial data labels. All the malicious devices share the same malicious dataset $D_M$ and $\pi_{t,i}$ is a batch of data samples from $D_M$ used to optimize the malicious objective. Except that they share a malicious dataset, the malicious attackers have the same background knowledge as the benign clients. The goal of the attackers is to make the global model achieve a good performance on the malicious objective (i.e. targeted misclassification on $D_M$). Considering the obliviousness of attack, the malicious devices also optimize benign objective, and the trade-off between the benign and malicious objectives is controlled by $\alpha$, where $\alpha \in [0, 1]$. Finally, the server averages the local models of the selected $K$ devices and updates the global model as follows:

$$W_t \leftarrow \frac{N}{K} \sum_{k \in \mathbb{S}_t} p^k W_{t,I}^k. \tag{4}$$

### 4.2  Estimation of Attack Effect on Model Parameters

Based on the above formulated training process, we analyze the impact of poisoning attacks on model parameters. To this end, we denote the set of attackers as $\mathbb{M}$, and introduce a new notation

$\boldsymbol{W}_t(\mathbb{S}_i \setminus \mathbb{M})$, which represents the global model weights in the $t$-th round when all malicious devices in $\mathbb{S}_i(i \leq t)$ do not perform the attack in the $i$-th training round. Specifically, when $i = t$, $\boldsymbol{W}_t(\mathbb{S}_t \setminus \mathbb{M})$ is optimized following:

$$\boldsymbol{W}_t(\mathbb{S}_t \setminus \mathbb{M}) \leftarrow \frac{N}{K} \sum_{k \in \mathbb{S}_t} p^k \boldsymbol{W}_{t,I}^k(\alpha = 1), \tag{5}$$

where $\boldsymbol{W}_{t,I}^k(\alpha = 1)$ indicates that $\boldsymbol{W}_{t,I}^k$ is trained using Equation 3 with setting $\alpha = 1$ (i.e., the $k$-th device is benign). A special case is $\boldsymbol{W}_t(\mathbb{S} \setminus \mathbb{M})$, which means the global model is optimized when all the malicious devices do not conduct attacks before the $t$-th round. To quantify the attack effect on the global model, we define the Attack Effect on Parameter (AEP) as follows:

**Definition 1.** *Attack Effect on Parameter (AEP), which is denoted as $\delta_t$, is the change of the global model parameters accumulated until $t$-th round due to the attack conducted by the malicious devices in the FL system:*

$$\delta_t \triangleq \boldsymbol{W}_t(\mathbb{S} \setminus \mathbb{M}) - \boldsymbol{W}_t. \tag{6}$$

Based on $AEP$, we can quantitatively evaluate the attack effect on the malicious objective using $F_M(\boldsymbol{W}_t(\mathbb{S} \setminus \mathbb{M}) - \delta_t) - F_M(\boldsymbol{W}_t(\mathbb{S} \setminus \mathbb{M}))$. As Figure 2 illustrates, although $\boldsymbol{W}_t(\mathbb{S} \setminus \mathbb{M})$ keeps updating after an adversarial round and there are no more attacks before the next adversarial round, the attack effect on the global model, i.e., $F_M$, remains for a number of rounds. Based on such an observation, we assume that the optimization of malicious objective is dominated by $\delta_t$ compared to $\boldsymbol{W}_t(\mathbb{S} \setminus \mathbb{M})$, which is learned from the benign objective. Consequently, if the attack effect in round $\tau$ remains for further rounds, $\|\delta_{t+1} - \delta_t\|$ should be small for $t \geq \tau$.

To analyze why the attack effect can persist in the global model, we consider the scenario where the malicious devices are selected in round $\tau_1$ and $\tau_2$, but will not be selected between these two rounds. We derive an estimator of $\delta_t$ for $\tau_1 < t < \tau_2$, denoted as $\hat{\delta}_t$:

$$\hat{\delta}_t = \frac{N}{K} \left[ \sum_{k \in \mathbb{S}_t} p^k \prod_{i=0}^{I-1} (\boldsymbol{I} - \eta_{t,i} \boldsymbol{H}_{t,i}^k) \right] \hat{\delta}_{t-1}, \tag{7}$$

where $\boldsymbol{H}_{t,i}^k \triangleq \nabla^2 F^k(\boldsymbol{W}_{t,i}^k, \xi_{t,i}^k)$. The derivation process is presented in Appendix D. Note that, we do not restrict the detailed malicious objective during derivation, and thus our estimator and analysis can be extended to other attacks, such as backdoor attacks.

## 4.3 Unveil Long-lasting Attack Effect

The key observation from Equation 7 is that if $\hat{\delta}_\tau$ is in the kernel of each $\boldsymbol{H}_{t,i}^k$ for $i$-th iteration where $k \in \mathbb{S}_t$ and $t > \tau$, then $\hat{\delta}_t$ will be the same as $\hat{\delta}_\tau$, which keeps $AEP$ in the global model. Based on this observation, we discover that **the reason why attack effects remain in the aggregated model is that the $AEP$s reside in the kernel of $\boldsymbol{H}_{t,i}^k$.** To validate our analysis, we conduct experiments on Fashion-MNIST with model poisoning attacks in FL. The experiment details and results are shown in Appendix B. The results show that $\|\boldsymbol{H}_{t,i}^k \delta_t\|_2$ would be nearly 0 under effective attacks. We also implement attack boosting by regularizing $\delta_t$ to be in the kernel of $\boldsymbol{H}_{t,i}^k$.

The above theoretical analysis and experiment results suggest that all the server-based defense methods (e.g. robust aggregation) will not be able efficiently mitigate the impact of model poisoning attacks to the victim global model. The fundamental reason for the failure of these mitigations is that: **the transmission of $AEP$ $\delta_t$ in global model is determined by $\boldsymbol{H}_{t,i}^k$, which is inaccessible by the central server.** Therefore, it is necessary to design an effective defense mechanism at client side aiming at mitigating attack that has already polluted the global model to further enhance the robustness of FL.

## 5 FL-WBC

### 5.1 Defense Design

Our aforementioned analysis shows that $AEP$ resides in the kernels of the Hessian matrices that are generated during the benign devices' local training. In this section, we propose *White Blood Cell*

*for Federated Learning (FL-WBC)* to efficiently mitigate the attack effect on the global model. In particular, we reform the local model training of benign devices to achieve two goals:

- Goal 1: To maintain the benign task's performance, loss of local benign task should be minimized.
- Goal 2: To prevent $AEP$ from being hidden in the kernels of Hessian matrices on benign devices, the kernel of $\boldsymbol{H}_{t+1,i}^k$ should be perturbed.

It is computationally unaffordable to perform the perturbance on $\boldsymbol{H}_{t,i}^k$ directly due to its high dimension. Therefore, in order to achieve Goal 2, we consider the essence of $\boldsymbol{H}_{t,i}^k$, i.e., second-order partial derivatives of the loss function, where the diagonal elements describe the change of gradients $\nabla F^k(\boldsymbol{W}_{t,i+1}^k) - \nabla F^k(\boldsymbol{W}_{t,i}^k)$ across iterations. We assume a fixed learning rate is applied for each communication round, and then $\nabla F^k(\boldsymbol{W}_{t,i+1}^k) - \nabla F^k(\boldsymbol{W}_{t,i}^k)$ can be approximated by $(\Delta \boldsymbol{W}_{t,i+1}^k - \Delta \boldsymbol{W}_{t,i}^k)/\eta_{t,i}$. In the experiments presented in §4.3, we observe that $\boldsymbol{H}_{t,i}^k$ has more than 60% elements to be zero in the most of iterations. When $\boldsymbol{H}_{t,i}^k$ is highly sparse, we add noise to the small-magnitude elements on its diagonal, which is approximately $(\Delta \boldsymbol{W}_{t,i+1}^k - \Delta \boldsymbol{W}_{t,i}^k)/\eta_{t,i}$, to perturb the null space of $\boldsymbol{H}_{t,i}^k$. Formally, we have two steps to optimize $\boldsymbol{W}_{t,i+1}^k$:

$$\boldsymbol{W}_{t,i+1}^{\hat{k}} = \boldsymbol{W}_{t,i}^k - \eta_{t,i}\nabla F^k(\boldsymbol{W}_{t,i}^k, \xi_{t,i}^k) \tag{8}$$

$$\boldsymbol{W}_{t,i+1}^k = \boldsymbol{W}_{t,i+1}^{\hat{k}} + \eta_{t,i}\Upsilon_{t,i}^k \odot \boldsymbol{M}_{t,i}^k, \tag{9}$$

where $\Upsilon_{t,i}^k$ is a matrix with the same shape of $\boldsymbol{W}$, and $\boldsymbol{M}_{t,i}^k$ is a binary mask whose elements are determined as:

$$\boldsymbol{M}_{t,i_{r,c}}^k = \begin{cases} 1, |(\boldsymbol{W}_{t,i+1}^{\hat{k}} - \boldsymbol{W}_{t,i}^k) - \Delta \boldsymbol{W}_{t,i}^k|_{r,c}/\eta_{t,i} \le |\Upsilon_{t,i_{r,c}}^k| \\ 0, |(\boldsymbol{W}_{t,i+1}^{\hat{k}} - \boldsymbol{W}_{t,i}^k) - \Delta \boldsymbol{W}_{t,i}^k|_{r,c}/\eta_{t,i} > |\Upsilon_{t,i_{r,c}}^k|, \end{cases} \tag{10}$$

where $\boldsymbol{M}_{t,i_{r,c}}^k$ is the element on the $r$-th row and $c$-th column of $\boldsymbol{M}_{t,i}^k$. Conceptually, $\boldsymbol{M}_{t,i+1}^k$ finds the small-magnitude elements on the $\boldsymbol{H}_{t,i}^k$'s diagonal.

Note that we have different choices of $\Upsilon_{t,i}^k$. In this work, we set $\Upsilon_{t,i}^k$ as Laplace noise with $mean = 0$ and $std = s$, since the randomness of $\Upsilon_{t,i}^k$ will make attackers harder to determine the defense strategy. Specifically, our defense is to find the elements in $\boldsymbol{W}_{t,i+1}^{\hat{k}}$ whose corresponding values in $|(\boldsymbol{W}_{t,i+1}^{\hat{k}} - \boldsymbol{W}_{t,i}^k) - \Delta \boldsymbol{W}_{t,i}^k|/\eta_{t,i}$ are smaller than the counterparts in $|\Upsilon_{t,i}^k|$. The detailed algorithm describing the local training process on benign devices when applying FL-WBC can be found in Appendix A. We derive a certified robustness guarantee for our defense, which provides a lower bound of distance of AEP between the adversarial round and the subsequent rounds. The detailed theorem of the certified robustness guarantee can be found in Appendix E.

## 5.2 Robustness to Adaptive attacks

Our defense is robust against adaptive attacks [25, 26] since the attacker cannot know the detailed defensive operations even after conducting the attack for three reasons. First, our defense is performed during the local training at the client side, where the detailed defensive process is closely related to benign clients' data. Such data is inaccessible to the attackers, and hence the attackers cannot figure out the detailed defense process. Second, even if the attackers have access to benign clients' data (which is a super strong assumption and beyond our threat model), the attackers cannot predict which benign clients will be sampled by the server to participate in the next communication round. Third, in the most extreme case where attackers have access to benign clients' data and can predict which clients will be sampled in the next round (which is an unrealistic assumption), the attackers still cannot successfully bypass our defense. The reason is that the defense during the benign local training is mainly dominated by the random matrix $\Upsilon_{t,i}^k$ in Equation 9, which is also unpredictable. With such unpredictability and randomness of our defense, no effective attack can be adapted.

# 6 Convergence Guarantee

In this section, we derive the convergence guarantee of FedAvg [1]—the most popular FL algorithm, with our proposed FL-WBC. We follow the notations in §4 describing FedAvg, and the only difference after applying FL-WBC is the local training process of benign devices. Specifically, for the $t$-th round, the local model on the $k$-th benign device is updated as:

$$\nabla F^{k'}(\boldsymbol{W}_{t,i}^k, \xi_{t,i}^k) = \nabla F^k(\boldsymbol{W}_{t,i}^k, \xi_{t,i}^k) + \mathcal{T}_{t,i} \tag{11}$$

$$\boldsymbol{W}_{t,i+1}^k \leftarrow \boldsymbol{W}_{t,i}^k - \eta_{t,i}\nabla F^{k'}(\boldsymbol{W}_{t,i}^k, \xi_{t,i}^k), \tag{12}$$

where $\mathcal{T}_{t,i}$ is the local updates generated by the perturbance step in Equation 9.

Our convergence analysis is inspired by [27]. Before presenting our theoretical results, we first make the following Assumptions 1-4 same as [27].

**Assumption 1.** $F^1, F^2, ..., F^N$ are $L$-smooth: $\forall \boldsymbol{V}, \boldsymbol{W}, F^k(\boldsymbol{V}) \leq F^k(\boldsymbol{W}) + (\boldsymbol{V}-\boldsymbol{W})^T\nabla F^k(\boldsymbol{W}) + \frac{L}{2}||\boldsymbol{V}-\boldsymbol{W}||_2^2$.

**Assumption 2.** $F_1, F_2, ..., F_N$ are $\mu$-strongly convex: $\forall \boldsymbol{V}, \boldsymbol{W}, F^k(\boldsymbol{V}) \geq F^k(\boldsymbol{W}) + (\boldsymbol{V}-\boldsymbol{W})^T\nabla F^k(\boldsymbol{W}) + \frac{\mu}{2}||\boldsymbol{V}-\boldsymbol{W}||_2^2$.

**Assumption 3.** Let $\xi_t^k$ be sampled from the $k$-th device's local data uniformly at random. The variance of stochastic gradients in each device is bounded: $\mathbb{E}||\nabla F^k(\boldsymbol{W}_{t,i}^k, \xi_{t,i}^k) - \nabla F^k(\boldsymbol{W}_{t,i}^k)||^2 \leq \sigma_k^2$ for $k = 1, ..., N$.

**Assumption 4.** The expected squared norm of stochastic gradients is uniformly bounded, i.e., $\mathbb{E}||\nabla F^k(\boldsymbol{W}_{t,i}^k, \xi_{t,i}^k)||^2 \leq G^2$ for all $k = 1, ..., N$, $i = 0, ..., I-1$ and $t = 0, ..., T-1$.

We define $F^*$ and $F^{k*}$ as the minimum value of $F$ and $F^k$ and let $\Gamma = F^* - \sum_{k=1}^{N} p_k F^{k*}$. We assume each device has $I$ local training iterations in each round and the total number of rounds is $T$. Then, we have the following convergence guarantee on FedAvg with our defense.

**Theorem 1.** Let Assumptions 1-4 hold and $L, \mu, \sigma_k, G$ be defined therein. Choose $\kappa = \frac{L}{\mu}$, $\gamma = \max\{8\kappa, I\}$ and the learning rate $\eta_{t,i} = \frac{2}{\mu(\gamma+tI+i)}$. Then FedAvg with our defense satisfies

$$\mathbb{E}[F(\boldsymbol{W}_T)] - F^* \leq \frac{2\kappa}{\gamma + TI}(\frac{Q+C}{\mu} + \frac{\mu\gamma}{2}\mathbb{E}||\boldsymbol{W}_0 - \boldsymbol{W}^*||^2),$$

where

$$Q = \sum_{k=1}^{N} p_k^2(s^2 + \sigma_k^2) + 6L\Gamma + 8(I-1)^2(s^2 + G^2)$$

$$C = \frac{4}{K}I^2(s^2 + G^2).$$

*Proof.* See our proof in Appendix F. □

# 7 Experiments

In our experiments, we evaluate FL-WBC against targeted model poisoning attack [11] described in §4 under both IID and non-IID settings. Experiments are conducted on a server with two Intel Xeon E5-2687W CPUs and four Nvidia TITAN RTX GPUs.

## 7.1 Experimental Setup

**Attack method.** We evaluate our defense against model poisoning attack shown in [11, 12]. There are several attackers in FL setup and all the attackers share a malicious dataset $D_M$, whose data points obey the same distribution with benign training data while having adversarial labels. We let all the attackers conduct the model poisoning attack at adversarial rounds $t_{adv}$ simultaneously such that the attack will be extremely strong.

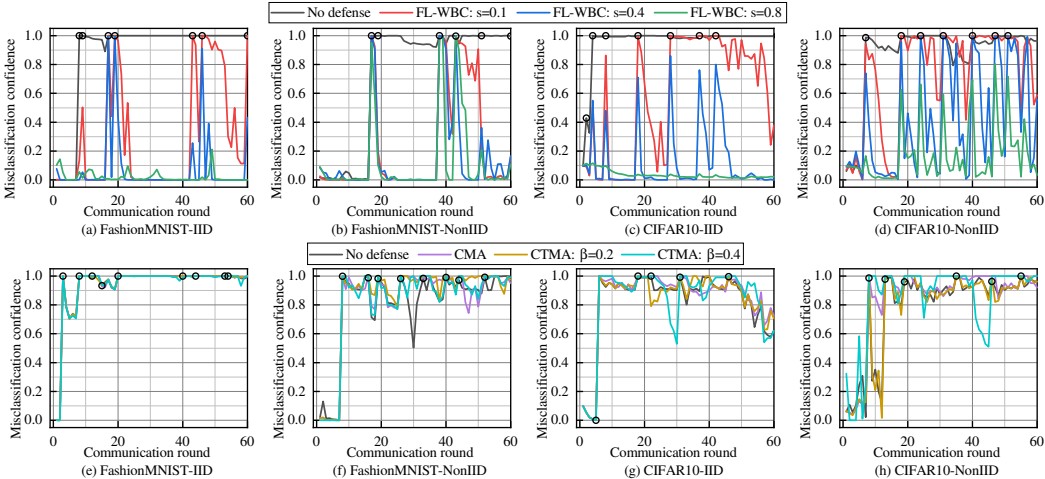

Figure 3: Comparison of misclassification confidence and communication round on FashionMNIST and CIFAR10 with IID/non-IID settings. The black circle denotes the adversarial rounds.

**Defense baseline.** We compare our proposed defense with two categories of defense methods that have been widely used: (1) **Differential privacy (DP)** improves robustness with theoretical guarantee by clipping the gradient norm and injecting perturbations to the gradients. We adopt both **Central Differential privacy (CDP)** [24] and **Local Differential privacy (LDP)** [24] for comparisons. We set the clipping norm as 5 and 10 for Fashion-MNIST and CIFAR10 respectively following [24] and apply Laplace noise with $mean = 0$ and $std = \sigma_{dp}$. (2) **Robust aggregation** improves robustness of FL by manipulating aggregation rules. We consider both **Coordinate Median Aggregation (CMA)** [13] and **Coordinate Trimmed-Mean Aggregation (CTMA)** [13] as baselines.

**Datasets.** To evaluate our defense under more realistic FL settings, we construct IID/non-IID datasets based on Fashion-MNIST and CIFAR10 by following the configurations in [1]. The detailed data preparation can be found in Appendix C. We sample 1 and 10 images from both datasets to construct the malicious dataset $D_M$ corresponding to scenarios $D_M$ having single image and multiple images. Note that, data samples in $D_M$ would not appear in training datasets of benign devices.

**Hyperparameter configurations.** Each communication round is set to be the adversarial round with probability 0.1. In each benign communication round, there are 10 benign devices which are randomly selected to participate in the training. In each adversarial round, 5 malicious and 5 randomly selected benign devices participate in the training, which means there are 50% attackers involved in adversarial rounds. Additional configurations and model structures can be found in Appendix C.

**Evaluation metrics.** (1) **Attack metric (misclassification confidence/accuracy:)** We define misclssification confidence/accuracy as the classification confidence/accuracy of the global model on the malicious dataset. (2) **Robust metric (attack mitigation rounds):** We define *attack mitigation rounds* as the number of communication rounds after which the misclassification confidence can decrease to lower than 50% or misclassification accuracy can decrease to lower than the error rate for the benign task. (3) **Utility metric (benign accuracy):** We use the accuracy of the global model on the benign testing set of the primary task to measure the effectiveness of FL algorithms (i.e., FedAvg [1]). The higher the accuracy is, the higher utility is obtained.

## 7.2 Effectiveness of FL-WBC with *Single* Image in The Malicious Dataset

We first show the results when there is only one image in the malicious dataset. We consider IID and non-IID settings for both Fashion-MNIST and CIFAR10 datasets. Figure 3 shows the misclassification confidence of our defense and the robust aggregation baselines in the first 60 communication rounds. The results show that our defense can more effectively and efficiently mitigate the impact of model poisoning attack in comparison with baseline methods. In particular, FL-WBC can mitigate the impact of model poisoning attack within 5 communication round when $s$ (i.e., standard deviation for $\Upsilon$) is 0.4 for both IID and non-IID settings. With regard to CMA and CTMA, the attack impact can not be mitigated within 10 subsequent rounds even when $\beta$ for CTMA is 0.4, where 80% of local

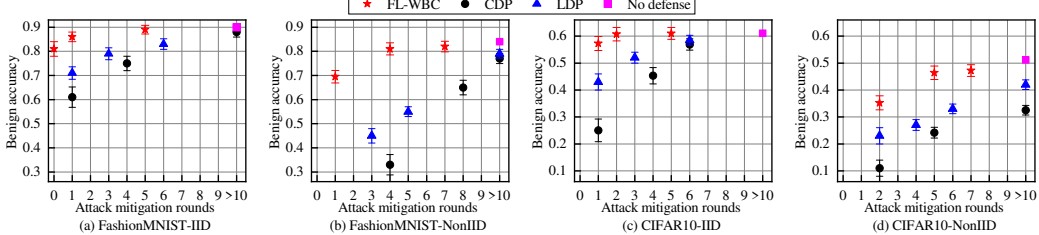

Figure 4: Comparison of benign accuracy and attack mitigation rounds on FashionMNIST and CIFAR10 with IID/non-IID settings when $D_M$ has only one image.

updates are trimmed before aggregation. Thus, the robust aggregation baselines fail to mitigate the model poisoning attack under our attack settings.

We also compare our defense with CDP and LDP in terms of *benign accuracy* and *attack mitigation rounds*. We evaluate our defense by varying $s$ from 0.1 to 1, and evaluate DP baselines by changing $\sigma_{dp}$ from 0.1 to 10. For each defense method, we show the trade-off between *benign accuracy* and *attack mitigation rounds* in Figure 4. We have two key observations: 1) With sacrificing less than 5% benign accuracy, FL-WBC can mitigate the impact of model poisoning attack on the global model in 1 communication round for IID settings, and within 5 communication rounds for non-IID settings respectively. However, CDP and LDP fail to mitigate attack effect within 5 rounds for IID and within 10 rounds for non-IID settings with less than 5% accuracy drop. 2) For non-IID settings where the defense becomes more challenging, FL-WBC can still mitigate the attack effect within 2 rounds with less than 15% benign accuracy drop, but DP can not make an effective mitigation within 3 rounds with less than 30% benign accuracy drop, leading to the unacceptable utility on the benign task. The reason of FL-WBC outperforming CDP and LDP is that FL-WBC only inject perturbations to the parameter space where the long-lasting $AEP$ resides in instead of perturbing all the parameters like DP methods. Therefore, FL-WBC can achieve better robustness with less accuracy drop.

In addition, we also observe that defense for non-IID settings is harder than IID settings, the reason is that under non-IID settings the devices train only a part of parameters [28] when holding only a few classes of data, leading to a sparser $H_{t,i}^k$ that is more likely to have a kernel with a higher dimension.

### 7.3 Effectiveness of FL-WBC with *Multiple* Images in The Malicious Dataset

We evaluate the defense effectiveness of robust aggregation baselines when $D_M$ has 10 images, and the results are shown in Table 1.

Table 1: Results of attack mitigation rounds for robust aggregations when $D_M$ has multiple images.

| Defense | Fashion-MNIST (IID) | Fashion-MNIST (non-IID) | CIFAR10 (IID) | CIFAR10 (non-IID) |
|---|---|---|---|---|
| CTMA ($\beta = 0.1$) | 7 | 9 | 8 | >10 |
| CTMA ($\beta = 0.2$) | 7 | 8 | 8 | 9 |
| CTMA ($\beta = 0.4$) | 6 | 8 | 7 | 9 |
| CMA | 5 | 7 | 6 | 8 |

Defense against the attack when $D_M$ has multiple images is easier than $D_M$ has only one image. The reason is that $AEP$ of multiple malicious images requires a larger parameter space to reside in compare to $AEP$ of single malicious image.

The results show that even though attack effect will be mitigated finally when there are multiple images in $D_M$, robust aggregation can not guarantee mitigating the attack effect within 5 communication rounds for both IID and non-IID settings.

We also evaluate the defense effectiveness of FL-WBC and DP baselines in terms of benign accuracy and attack mitigation rounds when $D_M$ has multiple images. The results are shown in Figure 5.

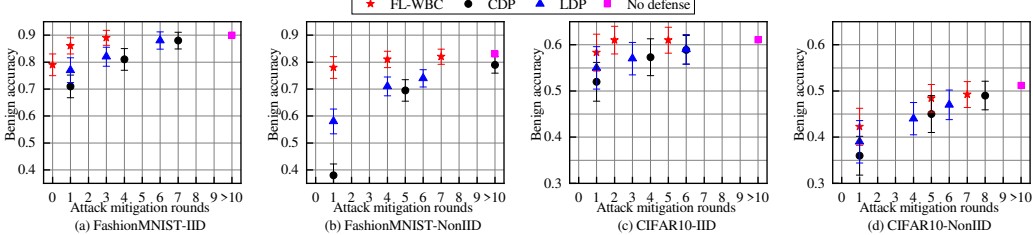

Figure 5: Comparison of benign accuracy and attack mitigation rounds on FashionMNIST and CIFAR10 with IID/non-IID settings when $D_M$ has multiple images.

The results show that FL-WBC can guarantee that attack impact will be mitigated in one round with sacrificing less than 3% benign accuracy for IID settings and 10% for non-IID settings, respectively. However, the DP methods incur more than 9% benign accuracy drop to achieve the same robustness for IID settings and 40% for non-IID settings, respectively. Therefore, FL-WBC significantly outperforms the DP methods in defending against model poisoning attacks.

## 7.4 Integration of The Robustness Aggregation and FL-WBC

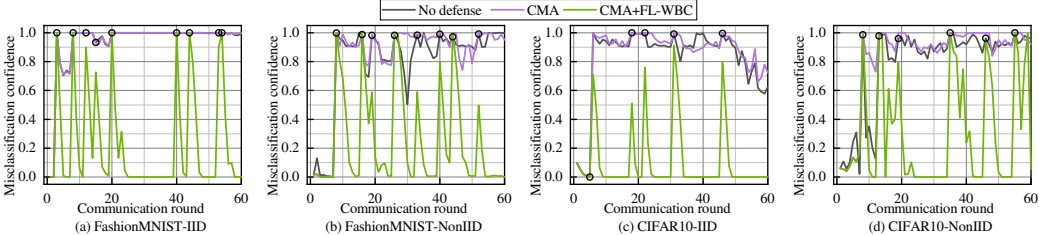

Figure 6: Comparison of misclassification confidence and communication round on FashionMNIST and CIFAR10 with IID/non-IID settings. The black circle denotes the adversarial rounds.

We also conduct experiments by integrating the robustness aggregation with FL-WBC to demonstrate that FL-WBC is complementary to server-based defenses. We conduct experiments by integrating Coordinate Median Aggregation (CMA) and FL-WBC. We set $s = 0.4$ for FL-WBC. After applying both CMA and FL-WBC with $s = 0.4$, the global model sacrifices less than 7% benign accuracy for both Fashion-MNIST and CIFAR10 dataset under IID/non-IID settings. We conduct experiments following the same setup in §7 with single image in the malicious dataset, and the results are shown in Figure 6.

The results show that only CMA can not mitigate the attack effect under our experimental setting. By applying both CMA and FL-WBC, the attack effect is mitigated within 1 communication rounds under IID settings and within 5 communication rounds under non-IID settings. Thus, our defense is complementary to the server-based robustness aggregations, and further enhance the robustness of FL against model poisoning attacks under extremely strong attacks.

## 8 Conclusion

We design a client-based defense against the model poisoning attack, targeting at the scenario where the attack that has already broken through the server-based defenses and polluted the global model. The experiment results demonstrate that our defense outperforms baselines in mitigating the attack effectively and efficiently, i.e., our defense successfully defends against the attack within fewer communication rounds with less model utility degradation. In this paper, we focus on the targeted poisoning attack [11, 12]. Our defense can be easily extended to many other poisoning attacks, such as backdoor attacks, since we do not restrict the malicious objective when deriving $AEP$.

## 9 Funding Transparency Statement

Funding in direct support of this work: NSF OIA-2040588, NSF CNS-1822085, NSF SPX-1725456, NSF IIS-2140247.

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
