# A Algorithm for training process applying FL-WBC

The detailed local training process of benign devcies after applying FL-WBC is shown in Algorithm 1. As Algorithm 1 shows, the only overheads after applying our defense is that devices need additional storage to store $\boldsymbol{W}_{-1}$ and $\boldsymbol{W}_{-2}$ during local training.

---

**Algorithm 1** Local training process applying FL-WBC on a benign device in round $t$.

---

**Input:** Local training data $\mathbb{D} \in \mathbb{R}^{L \times P \times Q}$; Local objective function $F : \mathbb{R}^{P \times Q} \to \mathbb{R}$; Local model parameters $\boldsymbol{W} \in \mathbb{R}^{M \times N}$; The number of local training iterations $I$; Learning rates $\eta_{t,i}$ for $i \in [I]$; Standard deviation of Laplace noise $s$.
**Output:** Learnt model parameter $\boldsymbol{W}$ with our defense.
 1: Initialize $\boldsymbol{W}_{-1}, \boldsymbol{W}_{-2}$;
 2: $i \leftarrow 0$;
 3: **for** batch $\mathcal{B}$ in $\mathbb{D}$ **do**
 4:     Randomly generate a Laplace noise matrix $\Upsilon \in \mathbb{R}^{M \times N}$ with $mean = 0$ and $std = s$;
 5:     $\boldsymbol{W}_{-1} \leftarrow \boldsymbol{W}$;
 6:     $\boldsymbol{W} \leftarrow \boldsymbol{W} - \eta_{t,i} \nabla F(\boldsymbol{W}, \mathcal{B})$;
 7:     **if** this is not the first training batch **then**
 8:         $\boldsymbol{W}^* \leftarrow (\boldsymbol{W} - \boldsymbol{W}_{-1}) - (\boldsymbol{W}_{-1} - \boldsymbol{W}_{-2})$;
 9:         Find the set $\mathbb{S}$ which contains the indices of elements in $|\boldsymbol{W}^*| - \eta_{t,i}|\Upsilon|$ which are less than or equal to 0;
10:         **for** $j, k \in \mathbb{S}$ **do**
11:             $\boldsymbol{W}_{j,k} \leftarrow \boldsymbol{W}_{j,k} + \eta_{t,i} \Upsilon_{j,k}$;
12:         **end for**
13:     **end if**
14:     $\boldsymbol{W}_{-2} \leftarrow \boldsymbol{W}_{-1}$;
15:     $i \leftarrow i + 1$
16: **end for**

---

# B Experiments to Support Analysis in §4.3

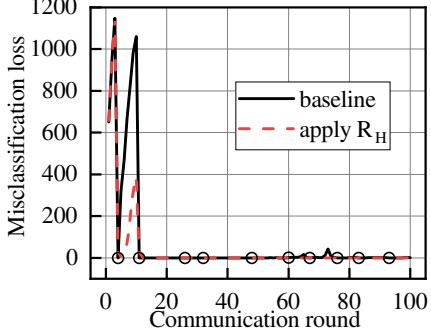

Figure 7: Compared results of misclassification loss on malicious dataset for model poisoning attack with and without applying $R_H(\boldsymbol{W})$. The black circle denotes the adversarial rounds.

To validate our analysis, we conduct experiments on Fashion-MNIST with model poisoning attacks in FL training. The training data is uniformly distributed to 100 devices (5 malicious devices included). In adversarial rounds, 5 malicious devices and 5 randomly chosen benign devices participate in training. In other rounds, 10 randomly selected benign devices participate in training. The model architecture can be found in Table 3 (Fashion-MNIST dataset). For training, we set local epoch $E$ as 1 and batch size $B$ as 32. We apply SGD optimizer and set the learning rate $\eta$ to 0.01.

We first compute $\mathbb{E}_{i \in [I], k \in \mathbb{S}_{t_{adv}+1}}(\boldsymbol{H}^k_{t_{adv}+1,i} \delta_{t_{adv}})$, denoted as $\Phi_{t_{adv}}$, for each adversarial round $t_{adv}$. In order to derive $\delta_{t_{adv}}$, the malicious devices in round $t_{adv}$ perform local training twice by setting $\alpha$ in Equation 3 as 1 and 0, respectively. Following that we have $\boldsymbol{W}_{t_{adv}}$ and $\boldsymbol{W}_{t_{adv}}(\mathbb{S}_{t_{adv}} \setminus \mathbb{M})$ respectively through aggregation. Then we derive $\delta_{t_{adv}}$ by computing the difference between $\boldsymbol{W}_{t_{adv}}(\mathbb{S}_{t_{adv}} \setminus \mathbb{M})$

| Adversarial round | baseline $\Phi_t$ | $\Phi_t$ applying $R_H(\boldsymbol{W})$ |
|---|---|---|
| 5 | 3.56 | 1.06 |
| 11 | 0.00 | 0.00 |
| 26 | 0.01 | 0.00 |
| 32 | 0.00 | 0.00 |
| 48 | 0.00 | 0.00 |
| 60 | 0.05 | 0.00 |
| 67 | 0.07 | 0.00 |
| 76 | 0.00 | 0.00 |
| 83 | 0.00 | 0.00 |
| 93 | 0.00 | 0.00 |

Table 2: Numerical results of $\Phi_t$ with and without applying $R_H(\boldsymbol{W})$ at adversarial rounds.

and $\boldsymbol{W}_{t_{adv}}$. Due to the extremely high dimension of $\boldsymbol{H}^k_{t_{adv}+1,i}$, we compute $\Phi_{t_{adv}}$ element by element following

$$\Phi_{t_{adv}\,m,n} = \mathbb{E}_{i\in[I],k\in\mathbb{S}_{t_{adv}+1}} \left\langle \nabla_W[\nabla_W F^k(\boldsymbol{W}^k_{t_{adv}+1,i}, \xi^k_{t_{adv}+1,i})_{m,n}]|\delta_{t_{adv}} \right\rangle \tag{13}$$

The results of baseline in Figure 7 shows the loss that the malicious data is miss-classified by the global model, and the computed $\Phi_t$ for each adversarial round is presented in Table 2. The results show that for the first adversarial round (i.e. round 5), the value of $\Phi_t$ is higher than 3 and the attack is mitigated rapidly. While for the following adversarial rounds, $\Phi_t$s are nearly 0, and the miss-classification loss keeps low until the next attack is conducted. This result is consistent with our analysis in §4 for why attack effects can remain in the global model.

In addition, we introduce a regularization $R_H(W)$ that approximates $\mathbb{E}_{i\in[I]}\|\boldsymbol{H}^k_{t_{adv},i}\delta_t\|_2$ into the local training of malicious devices, enforcing $\delta_t$ to reside in the null space of $\boldsymbol{H}^k_{t_{adv},i}$. Suppose $\boldsymbol{W}\in\mathbb{R}^{L\times M}$, then $R_H(W)$ for malicious devices $k$ is formulated as

$$R_H(\boldsymbol{W}) = \sum_{l\in[L],m\in[M]} \left\langle \nabla_W[\nabla_W F^k(\boldsymbol{W}^k_{t_{adv},0}, \xi^k_{t_{adv},0})_{l,m}]|\boldsymbol{W} - \boldsymbol{W}_{t_{adv}}(\mathbb{S}_{t_{adv}} \setminus \mathbb{M}) \right\rangle. \tag{14}$$

It is shown that we use the sum of elements in $\boldsymbol{H}^k_{t_{adv},0}\delta_t$ to approximate $\mathbb{E}_{i\in[I]}\|\boldsymbol{H}^k_{t_{adv},i}\delta_t\|_2$. We adopt this approximation due to unacceptable computational costs of computing $\mathbb{E}_{i\in[I]}\|\boldsymbol{H}^k_{t_{adv},i}\delta_t\|_2$ directly. When $\mathbb{E}_{i\in[I]}\|\boldsymbol{H}^k_{t_{adv},i}\delta_t\|_2$ is zero, the sum of elements in $\boldsymbol{H}^k_{t_{adv},0}\delta_t$ should also be zero. So $R_H(\boldsymbol{W})$ is a weaker regularization compared to $\mathbb{E}_{i\in[I]}\|\boldsymbol{H}^k_{t_{adv},i}\delta_t\|_2$.

We aim to evaluate whether the poisoning attack can be boosted after applying $R_H(\boldsymbol{W})$, i.e., the attack effect remains for more rounds. As Figure 7 and Table 2 show, the values of $\Phi_t$ in round 5, 60 and 67 are reduced after applying $R_H$ and the corresponding attacks are boosted. This result further supports our analysis that the reason why effective attacks can remain in the aggregated model is that the $AEP$s reside in the kernels of $\boldsymbol{H}^k_{t,i}$.

## C   Experiment setup

### C.1   Detailed data preparation and hyperparameter configurations for §7

For IID settings, the data is uniformly distributed to 100 devices (malicious devices included). For non-IID settings, we first sort the data by the digit label, divide it into 200 shards uniformly, and assign each of 100 clients (malicious devices included) 2 shards.

For training, we set local epoch $E$ as 1 and batch size $B$ as 32. We apply SGD optimizer and set the learning rate $\eta$ to 0.01. We set 5 devices out of totally 100 devices to be malicious. The model architectures for two dataset are shown in Table 3. We conduct 500 communication rounds of training for Fashion-MNIST and 1000 communication rounds for CIFAR10.

Table 3: Model architectures for *Fashion-MNIST* dataset and *CIFAR10* dataset.

| *Fashion-MNIST* | *CIFAR10* |
|---|---|
| $5 \times 5$ Conv 1-16 | $5 \times 5$ Conv 3-6 |
| $5 \times 5$ Conv 16-32 | $3 \times 3$ Maxpool |
| FC–10 | $5 \times 5$ Conv 6-16 |
| | $3 \times 3$ Maxpool |
| | FC–120 |
| | FC–84 |
| | FC–10 |

# D   Derivation of Equation 7

Since no malicious devices are selected between round $\tau_1$ and round $\tau_2$, the training processes of $\boldsymbol{W}_{t-1}(\mathbb{S} \setminus \mathbb{M})$ to $\boldsymbol{W}_t(\mathbb{S} \setminus \mathbb{M})$ and $\boldsymbol{W}_{t-1}$ to $\boldsymbol{W}_t$ are the same. The cause of difference between $\boldsymbol{W}_t(\mathbb{S} \setminus \mathbb{M})$ and $\boldsymbol{W}_t$ is that they are locally trained from different initial models, $\boldsymbol{W}_{t-1}(\mathbb{S} \setminus \mathbb{M})$ and $\boldsymbol{W}_{t-1}$, respectively.

We first estimate the term $\boldsymbol{W}_t^k(\mathbb{S} \setminus \mathbb{M}) - \boldsymbol{W}_t^k$ by applying first-order Taylor approximation and the chain rule based on Equation 2:

$$\boldsymbol{W}_{t,I}^k(\mathbb{S} \setminus \mathbb{M}) - \boldsymbol{W}_{t,I}^k$$
$$\approx \frac{\partial \boldsymbol{W}_{t,I}^k}{\partial \boldsymbol{W}_{t,I-1}^k} \frac{\partial \boldsymbol{W}_{t,I-1}^k}{\partial \boldsymbol{W}_{t,I-2}^k} \cdots \frac{\partial \boldsymbol{W}_{t,1}^k}{\partial \boldsymbol{W}_{t,0}^k}(\boldsymbol{W}_{t,0}^k(\mathbb{S} \setminus \mathbb{M}) - \boldsymbol{W}_{t,0}^k). \tag{15}$$

where $\boldsymbol{W}_{t,0}^k(\mathbb{S} \setminus \mathbb{M}) - \boldsymbol{W}_{t,0}^k = \boldsymbol{W}_{t-1}(\mathbb{S} \setminus \mathbb{M}) - \boldsymbol{W}_{t-1}$. According to Equation 2, we have

$$\frac{\partial \boldsymbol{W}_{t,i+1}^k}{\partial \boldsymbol{W}_{t,i}^k} = \boldsymbol{I} - \eta_{t,i} \boldsymbol{H}_{t,i}^k, \tag{16}$$

where $\boldsymbol{H}_{t,i}^k \triangleq \nabla^2 F^k(\boldsymbol{W}_{t,i}^k, \xi_{t,i}^k)$.

Afterwards, according to Equation 4, we obtain

$$\boldsymbol{W}_t(\mathbb{S} \setminus \mathbb{M}) - \boldsymbol{W}_t = \frac{N}{K} \sum_{k \in \mathbb{S}_t} p^k [\boldsymbol{W}_{t,I}^k(\mathbb{S} \setminus \mathbb{M}) - \boldsymbol{W}_{t,I}^k]. \tag{17}$$

By combining Equations 15, 16 and 17, we get an estimator of $\delta_t$, denoted as $\hat{\delta}_t$

$$\hat{\delta}_t = \frac{N}{K} [\sum_{k \in \mathbb{S}_t} p^k \prod_{i=0}^{I-1} (\boldsymbol{I} - \eta_{t,i} \boldsymbol{H}_{t,i}^k)] \hat{\delta}_{t-1}. \tag{18}$$

# E   Certified robustness guarantee

We define our *certified robustness guarantee* as the distance of $AEP$ between the adversarial round and the subsequent rounds where FL-WBC is applied. A larger distance of $AEP$ indicates that FL-WBC can mitigate the attack more efficiently. We first make an assumption for the Hessian matrix after applying our defense:

**Assumption 5.** *After applying FL-WBC, the new benign training Hessian matrix $H'$ would be nearly full-rank. Following the notation of $s$ as the variance of $\Upsilon$ in Equation 9, the operator $H'$ is bounded below: $\mathbb{E}\|H'\boldsymbol{a}\|_2^2/P \geq s\|\boldsymbol{a}\|_2^2$.*

Additionally, we make assumptions on bounding the norm of $AEP$s in different rounds and independent distributions of $H_{t,i}^k \delta_{t,i}^k$.

**Assumption 6.** *The expected norm of $AEP$s is lower bounded in each round across devices and iterations: $\mathbb{E}_{i \in [I], k \in \mathbb{S}_t} \|\delta_{t,i}^k\|_2^2 \geq \Lambda_t$.*

**Assumption 7.** *Since local training data have independent distributions, the elements in $H_{t,i}^k \delta_{t,i}^k$ are independent and have expectations of 0 in different rounds across devices and iterations.*

Following the notations in §4 and applying the local training of benign devices in Algorithm 1, which can be found in Appendix A, we have the following theorem for our *robustness guarantee* on FedAvg after applying FL-WBC.

**Theorem 2.** *Let Assumption 5-7 hold and $s, \Lambda_t, P$ be defined therein. $K$ denotes the number of devices involved in training for each round. Assuming that poisoning attack happens in round $t_{adv}$ and no attack happens from round $t_{adv}$ to $T$, then for FedAvg applying FL-WBC we have:*

$$\mathbb{E}\|\hat{\delta}_T - \hat{\delta}_{t_{adv}}\|_2^2 \geq \frac{PIs}{K} \sum_{t=t_{adv}+1}^{T} \eta_{t,I-1}^2 \Lambda_t. \tag{19}$$

*Proof.* According to Equations 15 and 16, we obtain

$$\hat{\delta}_{t,i+1}^k = (\boldsymbol{I} - \eta_{t,i}\boldsymbol{H}_{t,i}^k)\hat{\delta}_{t,i}^k. \tag{20}$$

According to Equation 17, we have

$$\mathbb{E}(\hat{\delta}_t) = \mathbb{E}(\frac{N}{K} \sum_{k \in \mathbb{S}_t} p^k \hat{\delta}_{t,I}^k). \tag{21}$$

With $\hat{\delta}_{t-1} = \hat{\delta}_{t,0}^k$ and $\mathbb{E}(p^k) = \frac{1}{N}$, we have

$$\mathbb{E}(\hat{\delta_{t-1}}) = \mathbb{E}(\frac{N}{K} \sum_{k \in \mathbb{S}_t} p^k \hat{\delta}_{t,0}^k). \tag{22}$$

Then we have

$$\mathbb{E}(\hat{\delta}_t - \hat{\delta}_{t-1}) = \mathbb{E}[\frac{N}{K} \sum_{k \in \mathbb{S}_t} p^k (\hat{\delta}_{t,I}^k - \hat{\delta}_{t,0}^k)] \tag{23}$$

$$= \mathbb{E}[\frac{N}{K} \sum_{k \in \mathbb{S}_t} p^k \sum_{i=0}^{I-1} (\hat{\delta}_{t,i+1}^k - \hat{\delta}_{t,i}^k)] \tag{24}$$

$$= \mathbb{E}[\frac{N}{K} \sum_{k \in \mathbb{S}_t} p^k \sum_{i=0}^{I-1} -\eta_{t,i}\boldsymbol{H}_{t,i}^k \hat{\delta}_{t,i}^k], \tag{25}$$

where the first equality comes from Equations 21 and 22, the third equality comes from Equation 20. Accumulating the difference between $\hat{\delta}_t$ and $\hat{\delta}_{t-1}$ formulated in Equation 25, we have

$$\mathbb{E}(\hat{\delta}_T - \hat{\delta}_{t_{adv}}) = \mathbb{E}[\sum_{t=t_{adv}+1}^{T} (\hat{\delta}_t - \hat{\delta}_{t-1})] \tag{26}$$

$$= \mathbb{E}[\sum_{t=t_{adv}+1}^{T} \sum_{k \in \mathbb{S}_t} \sum_{i=0}^{I-1} -\frac{N}{K} p^k \eta_{t,i} \boldsymbol{H}_{t,i}^k \hat{\delta}_{t,i}^k]. \tag{27}$$

Then we have

$$\mathbb{E}\|\hat{\delta}_T - \hat{\delta}_{t_{adv}}\|_2^2 = \mathbb{E}\| \sum_{t=t_{adv}+1}^{T} \sum_{k \in \mathbb{S}_t} \sum_{i=0}^{I-1} -\frac{N}{K}p^k \eta_{t,i} \boldsymbol{H}_{t,i}^k \hat{\delta}_{t,i}^k \|_2^2 \tag{28}$$

$$= \sum_{t=t_{adv}+1}^{T} \sum_{k \in \mathbb{S}_t} \sum_{i=0}^{I-1} \mathbb{E}\| \frac{N}{K}p^k \eta_{t,i} \boldsymbol{H}_{t,i}^k \hat{\delta}_{t,i}^k \|_2^2 \tag{29}$$

$$\geq \sum_{t=t_{adv}+1}^{T} \mathbb{E} \sum_{k \in \mathbb{S}_t} (\frac{N}{K}p^k \eta_{t,I-1})^2 \sum_{i=0}^{I-1} Ps\|\hat{\delta}_{t,i}^k\|_2^2 \tag{30}$$

$$\geq \sum_{t=t_{adv}+1}^{T} \mathbb{E} \sum_{k \in \mathbb{S}_t} (\frac{N}{K}p^k \eta_{t,I-1})^2 \sum_{i=0}^{I-1} Ps\Lambda_t \tag{31}$$

$$= \sum_{t=t_{adv}+1}^{T} \mathbb{E} \sum_{k \in \mathbb{S}_t} (\frac{\eta_{t,I-1}}{K})^2 IPs\Lambda_t \tag{32}$$

$$= \frac{PIs}{K} \sum_{t=t_{adv}+1}^{T} (\eta_{t,I-1})^2 \Lambda_t, \tag{33}$$

$$\tag{34}$$

where the second equality come from Assumption 7, the first inequality comes from the lower bound of operator $\boldsymbol{H}_{t,i}^k$ parameterized in Assumption 5 and shrinking learning rate, the second inequality comes form the bounded norm of AEP stated in Assumption 6. The third equality comes from the equation $\mathbb{E}(p^k) = \frac{1}{N}$.

$\square$

## F   Proof of Theorem 1

**Overview:** Our proof is mainly inspired by [27]. Specifically, our proof has two key parts. First, we derive the bounds similar to those in Assumptions 3 and 4, after applying our defense scheme. Second, we adapt **Theorem 2** on convergence guarantee in [27] using our new bounds.

**Bounding the expected distance between the perturbed gradients with our defense and raw gradients.** According to Algorithm 1, the absolute values of elements in $\mathcal{T}_{t,i}$ in Equation 11 is $\max\{|\Upsilon| - |\boldsymbol{W}^*|/\eta_{t,i}, 0\}$. Thus, $\|\mathcal{T}_{t,i}\|_2^2 \leq \|\Upsilon\|_2^2$. Then we have

$$\mathbb{E}||\nabla F_k'(\boldsymbol{W}_{t,i}^k, \xi_{t,i}^k) - \nabla F_k(\boldsymbol{W}_{t,i}^k, \xi_{t,i}^k)||_2^2 \tag{35}$$

$$=\mathbb{E}||\mathcal{T}_{t,i}||_2^2 \leq \mathbb{E}||\Upsilon||_2^2 = \mathbb{E}(\Upsilon)^2 + Var(\Upsilon) = s^2. \tag{36}$$

**New bounds for Assumption 3 with our defense.**

We use the norm triangle inequality to bound he variance of stochastic gradients in each device, and we have

$$\mathbb{E}||\nabla F_k'(\boldsymbol{W}_{t,i}^k, \xi_{t,i}^k) - \nabla F_k(\boldsymbol{W}_{t,i}^k)||^2 \tag{37}$$

$$\leq \mathbb{E}||\nabla F_k'(\boldsymbol{W}_{t,i}^k, \xi_{t,i}^k) - \nabla F_k(\boldsymbol{W}_{t,i}^k, \xi_{t,i}^k)||^2 \tag{38}$$

$$+ \mathbb{E}||\nabla F_k(\boldsymbol{W}_{t,i}^k, \xi_{t,i}^k) - \nabla F_k(\boldsymbol{W}_{t,i}^k)||^2 \tag{39}$$

$$\leq s^2 + \sigma_k^2, \tag{40}$$

where we use Assumption 3 and Equation 36 in Equation 40.

**New bounds for Assumption 4 with our defense.** The expected squared norm of stochastic gradients $\nabla F_k'(\boldsymbol{W}_{t,i}^k, \xi_{t,i}^k)$ with our defense is as follows:

$$\mathbb{E}||\nabla F_k'(\boldsymbol{W}_{t,i}^k, \xi_{t,i}^k)||^2 \tag{41}$$

$$\leq \mathbb{E}||\nabla F_k'(\boldsymbol{W}_{t,i}^k, \xi_{t,i}^k) - \nabla F_k(\boldsymbol{W}_{t,i}^k, \xi_{t,i}^k)||^2 \tag{42}$$

$$+ \mathbb{E}||\nabla F_k(\boldsymbol{W}_{t,i}^k, \xi_{t,i}^k)||^2 \tag{43}$$

$$\leq s^2 + G^2, \tag{44}$$

where we use Assumption 4 and and Equation 36 in Equation 44.

**Convergence guarantee for FedAvg with our defense.** We define $F^*$ and $F_k^*$ as the minimum value of $F$ and $F_k$ and let $\Gamma = F^* - \sum_{k=1}^{N} p_k F_k^*$. We assume each device has $I$ local training iterations in each round and the total number of rounds is $T$. Let Assumptions 1-4 hold and $L, \mu, \sigma_k, G$ be defined therein. Choose $\kappa = \frac{L}{\mu}$, $\gamma = \max\{8\kappa, I\}$ and the learning rate $\eta_{t,i} = \frac{2}{\mu(\gamma+tI+i)}$.

By applying our new bounds and **Theorem 2** in [27], FedAvg using our defense has the following convergence guarantee:

$$\mathbb{E}[F(\boldsymbol{W}_T)] - F^* \leq \frac{2\kappa}{\gamma + TI}(\frac{Q+C}{\mu} + \frac{\mu\gamma}{2}\mathbb{E}||\boldsymbol{W}_0 - \boldsymbol{W}^*||^2),$$

where

$$Q = \sum_{k=1}^{N} p_k^2(s^2 + \sigma_k^2) + 6L\Gamma + 8(I-1)^2(s^2 + G^2)$$

$$C = \frac{4}{K}I^2(s^2 + G^2).$$