# OpenReview forum: "FL-WBC: Enhancing Robustness against Model Poisoning Attacks in Federated Learning from a Client Perspective"
_NeurIPS.cc/2021/Conference — NeurIPS 2021 Poster_

### Official Review · Reviewer_fBKQ · 2021-07-02

**Rating:** 7
**Confidence:** 3

**Summary:**

This paper proposes a client-based defence in federated learning against attacks which have already broken through server-side defences and which would otherwise persist through subsequent rounds after the adversarial attack has taken place. They place, under certain assumptions, theoretical bounds on the effectiveness of their approach and on its convergence. Experiments support the approach used.

**Limitations And Societal Impact:**

The authors do not specifically address potential societal impacts.

**Main Review:**

This is a very clearly written paper. Though not an expert in this area, the key idea (that attacks persist as their effect on the parameters lies in the kernel of certain Hessians) is plausible and apparently novel. Experimental evidence is used to bolster this hypothesis too.

This work seems significant in that it offers a seemingly robust (and somewhat quantifiable, based on their theoretical contributions) defence, which importantly can be used in conjunction with server-side defences.

The theoretical results require a large number of assumptions for their proof, and as a non-expert I am not confident with regard to how likely these assumptions are to be met. That said, some of the assumptions match those from previous work. Also, a quantitative theoretical approach is to be applauded – though the measure AEP seems overly simplistic.

Though the key argument feels consistent, i,e. that if the change in the parameters lies in the Hessians, it will persist, this would seem a sufficient condition though not a necessary one for an attack to persist from round to round.  In this vein, the “certified robustness” guarantee does not really seem to be a guarantee of robustness, but rather that this particular sufficient condition for attack persistence may not be met.

The experiments seem quite thorough and assess performance against SOTA of the defence and also impact on utility.

Typos/miscellaneous comments:
-	“Unfortunately, Often...”
-	I am uncomfortable with the notation $W_t(S_i \setminus M)$. It is not really parameterised by the indicated set (if the sets for i=1 and i=2 matched the parameters would match but these are different cases), but it is more that $i$ should be the parameter, as well as some indication of whether or not malicious devices are included.
-	Equation (7) has a leading N/K in its derivation in the Appendix
-	“equivalent to minimize the dimension...” should be “minimizing”


**Time Spent Reviewing:**

3

---

> ### Author Response · Authors · 2021-08-10
> **Response to Reviewer fBKQ**
>
> Thank you for your positive response and valuable comments.
>
> **Assumptions and simple AEP measure** Thanks a lot for appreciating our theoretical work. We follow previous papers [26,28]  to make the assumptions, which are common in theoretical analysis for FL. Our AEP measurement is simple yet effective, and its effectiveness has been demonstrated by our evaluations in Appendix B. Our defense based on our AEP measurement is also evaluated to be effective in Sec 7.
>
> **Sufficient rather not necessary condition** We appreciate your insight of our paper. It is true that AEP lying in the kernel of Hessian matrices is a sufficient rather than necessary condition for the attack effect to be long-lasting and it is hard to theoretically prove whether this is a necessary condition. However, we empirically show that persisting AEP in the kernel of Hessian matrices is truly the essential cause of long-lasting attack using the experiments in Appendix B. Even though there might be other causes of long-lasting attack, the certified robustness guarantee we derived in this paper focuses on the attack effect caused by the persisting AEP residing in the kernel of Hessian matrices. Thanks for your suggestion and we will clarify this point more clearly in our final version.
>
>
>
> **Typos/miscellaneous comments**
>
> (1) \& (3) \& (4) Thanks for pointing out the typos. We have fixed them in our new version.
>
> (2) Thanks for your detailed comments about our notation. We plan to reorganize previous $W_t(S_i \setminus M)$ into $\hat{W}_{t,\{i\}}$, such that it is parameterised directly by $i$ rather than the set $S_i$. What do you think of this new notation?
>
>
> [28] Sun, J., Li, A., Wang, B., Yang, H., Li, H., & Chen, Y. (2020). Provable Defense against Privacy Leakage in Federated Learning from Representation Perspective. arXiv preprint arXiv:2012.06043.

---

> > ### Comment · Reviewer_fBKQ · 2021-08-19
> > **Replying to authors**
> >
> > Dear authors,
> >
> > Thank you for your responses to my comments and for undertaking to address them as possible.
> >
> > I retain my positive rating of the paper unchanged.

---

### Official Review · Reviewer_HBHN · 2021-07-09

**Rating:** 7
**Confidence:** 3

**Summary:**

This paper shows that existing "robust aggregation" defenses against poisoning attacks in federated learning are ineffective in presence of strong attacks. Hence, they propose a client-based defense named "White Blood Cell for Federated Learning (FL-WBC), which can mitigate against attacks who already managed to pollute the global model.

**Ethical Concerns:**

No ethical concerns.

**Limitations And Societal Impact:**

I feel there is not enough discussion about possible limitations of the work, or at least of the experimental evaluation. I guess that these aspects should be taken into account:
* generalization of the results from the computer vision task to other domains; moreover, the specific focus on Fashion-MNIST and CIFAR-10 and not larger CV datasets may be affecting generalization
* the amount of adversarial rounds considered in the evaluation
* adaptive attackers that know the defense


**Main Review:**

# Strengths
- Their client-based defense works also in presence of already-polluted models, and is complimentary to server-based robust aggregation approaches
- They achieve convergence guarantees and certified robustness guarantees
- Author consider both IID and non-IID settings
- They demonstrate ineffectiveness of existing defenses against stronger attacks
- Code and datasets are included

# Weaknesses
- Possible lack of adaptive attackers
- Only tested on FashionMNIST and CIFAR-10 (somewhat simple datasets)
- There are many relevant information relegated to the Appendix (e.g., description of Figure 2, non-iid settings,  relevance of AEPs in Fashion-MNIST).

# Detailed comments

The paper is generally very well written, and positions itself very clearly with respect to the state of the art. Both the empirical and theoretical foundations of the work are extremely strong, but it is also clear what is the high-level intuition behind the proposed approach, so that the reader always know where they are despite the highly technical content of the paper.

**Presentation**. Despite the overall great writing style, I feel that there are just too many important information relegated to the Appendix, to the point that it is occasionally hard to assess the paper as-is without having a look at the Appendix. I feel that the writing and presentation should be somehow revised to present this. Some major issues I've had in terms of missing information from the main test include:
- information for Figure 2 (beta is not explained)
-  he high-level intuition non-iid settings,
- the experimental results on AEPs relevance in Fashion-MNIST
Moreover, the paper is very dense and full of symbols. While most of them are fully aligned with the state of the art, I feel especially for AEP-related symbols (e.g., "s", "H") it would be good to have a "symbol table" to improve readability of the work.

**Adaptive attackers**. In the model considered throughout the paper and depicted in Figure 1, I think that one thing that is not fully considered is what happens if malicious clients know that FL-WBC is being used. Is there an adaptive, stronger attack that could take advantage of the new component of the loss function, to create stronger and more subtle attacks? If it is only discussed in the text, it should be highlighted better, but it would be appropriate to have a specific section on adaptive attackers. See the following papers for reference:
- Tramer, Florian, et al. "On adaptive attacks to adversarial example defenses." arXiv preprint arXiv:2002.08347 (2020).
- Carlini, Nicholas, et al. "On evaluating adversarial robustness." arXiv preprint arXiv:1902.06705 (2019).

**Adversarial round probability**. How strong is the importance of the adversarial round probability as part of the experimental evaluation? I feel that from Figure 3 the number of adversarial rounds remains fairly limited. The authors mention approaches for 'detecting' that an attack is going on, but the whole paper is framed around _super strong adversaries_, so I was a bit confused when I saw such a low probability in the experimental evaluation.

**Generalization**. Is there any chance that since you are using relatively simple datasets such as MNIST and CIFAR-10 there is some artifacts in the actual robustness achieved?

# Minor comments
- Figure 4: Why the 'no defense' bullet is only for attack mitigation >10?
- there is a capital "Often" on page 3, which should be lowercase
- At the end of page 4, within the explanation after Equation (5):  when $\alpha=1$ the $k$-th device should be benign, right?

**Time Spent Reviewing:**

6

---

> ### Author Response · Authors · 2021-08-10
> **Response to Reviewer HBHN**
>
> Thank you for your positive response and valuable comments.
>
> **Presentation** We will explain $\beta$ in Figure 2 in Sec 3 and introduce the high-level intuition of non-iid settings in Sec 7.1 in the final version. For the evaluations of AEPs relevance on Fashion-MNIST, we will present more detailed results in Sec 4.3.
> Since we provide very detailed theoretical analysis, it is unavoidable the a large number of symbols appear in our paper. We think your suggestion of a symbol table is a great idea, and we will add it in Appendix.
>
> **Adaptive attackers** We actually described in line 233-235 that the configurations of our defense strategy are hard to be determined by the attacker. However, we think it is a good idea to have a specific section discussing why our defense is not vulnerable to adaptive attackers, and we want to clarify the key reasons here. Some defenses are vulnerable to adaptive attacks because their  defensive operations are determined even before the attack is conducted. For example, when applying *Norm thresholding of updates*[7], the attacker knows the exact details of the defensive operation, i.e., the local updates with larger significance will be cropped. As such, an adaptive attacker can hide the attack effect in gradients with smaller significance by regularizing malicious updates. However, in our defense, the attacker cannot know the detailed defensive operation even after conducting the attack for three reasons: (1) Our defense is on the client side, and the defense happens during the local training after attacks. Since our detailed defensive process is closely related to other benign clients' data, which is inaccessible to the attackers, the attackers cannot determine the detailed defense process. (2) Even though the attackers have access to benign clients' data (which is a super strong assumption and beyond our threat model), the attackers cannot predict which benign clients will be sampled by the server to participate in training in the next communication round. (3) In the most extreme case where attackers have access to benign clients' data and can predict which clients will be sampled in the next round (which is an unrealistic assumption), the attacker still cannot successfully bypass our defense. The reason is that the defense during the benign local training is mainly dominated by the random noise $\Upsilon$ in Equ (13), which is also unpredictable. Such unpredictability and randomness make our defense strategy robust to the attackers, such that no effective attack can be adapted. In addition, no matter what adaptation the attacker made, the AEP still needs model parameter subspace to reside in. Since our defensive goal is to zero the dimension of the subspace that long-lasting AEP could reside in, any attack adaptation is hard to be effective. We will add an additional section to include the above clarifications regarding the adaptive attacker in the final version.
>
> **Adversarial round probability** Thanks for your careful review and valuable thoughts! The strength of an attack can be described from two perspectives: **intensity** (i.e. how many attackers are involved in one time attack) and **frequency** (i.e. how frequently the attacks could happen in FL, which is also the adversarial round probability). For *super strong adversaries* in this paper, we focus on the attacks with high intensity (i.e. half of the clients in adversary rounds are attackers such that the server-side defenses will be  ineffective). However, it does not mean that our defense will be less effective when adversarial round probability goes higher. We do not set adversarial round probability in evaluation to be too high because we want to show the comparison of attack mitigation process between our defense and baselines more completely. In general, as long as the attack interval is not shorter than the communication rounds our defense needs to mitigate the attack (which are rather few), the attack effect will not be accumulated after applying our defense.
>
> **Generalization** Currently we only use Fashion-MNIST and CIFAR10 because they are both benchmark datasets in FL. We believe that our defense can achieve similar robustness on other dataset since we derive a theoretical robustness guarantee. We will still conduct evaluations on other more complicated datasets in the future work.
>
> **Minor comments**
>
> (1) The 'no defense' bullet is only for attack mitigation rounds >10 because the attack effect can not be mitigated within 10 communication rounds without defense.
>
> (2) Thanks for pointing out the typo. We have fixed it in our new version.
>
> (3) Yes, when $\alpha=1$, the $k$-th device should be benign. We have fixed this typo in our new version.

---

> > ### Comment · Reviewer_HBHN · 2021-08-18
> > **Replying to authors**
> >
> > Dear authors,
> >
> > thank you for your clarifications. As you mention in your comment, it'd be great if you could integrate these clarifications into text (even in reduced form, I am mindful of space).
> >
> > I keep my positive opinion of the paper.

---

### Official Review · Reviewer_ZHqh · 2021-07-16

**Rating:** 5
**Confidence:** 4

**Summary:**

The paper proposes White Blood Cell for Federated learning (FL-WBC) -- a client-based defense that can mitigate model poisoning attacks that have already polluted the global model.
The authors claim that strong model poisoning attacks that can circumvent server-based defenses can continue to impact the global model even if there are no subsequent attacks. Towards this end, the paper identifies the attack effect on the parameter space (AEP) and observes that the parameter subspace used for the attack is both inaccessible to the server and remains hidden in the kernel of the Hessian matrices on benign agents. Thus, they propose a client-based optimization that is designed to minimize the loss on the benign task while also minimizing the dimensionality of the Hessian kernel. Robustness certificate and convergence guarantees are also provided.
Evaluation is performed on FashionMNIST and CIFAR-10 datasets to demonstrate the ability of the defense to mitigate attacks using three metrics (attack metric, robust metric, and utility metric). Comparison is performed with DP-based defenses, CMA, and CTMA.

**Limitations And Societal Impact:**

Limitations and societal impact are not currently mentioned but should be added as either a separate discussion section or as part of the conclusion.

**Main Review:**

I have the following concerns with the defense:
a. Prior papers in this area including [8, 10,  11, 12] observe that the strength of an attack (even without a defense) starts to attenuate if there is no reinforcement by attackers in subsequent rounds. The benign updates are able to overwrite the effect of the attack. This is both intuitive and has been experimentally validated in the past. I really like the AEP analysis performed by the authors but I am not fully convinced, especially as it does not seem to reconcile with prior observations.

b.  Second, the evaluation does not consider an adaptive attacker that is aware of the client-based optimization performed to remove the effect on the parameter subspace (hiding the attack).

c. The authors need to explicitly mention the threat model which will then provide the parameters for attacks that can be mounted on the defense.  Furthermore, the solution requires clients participating in federated learning to perform a specific form of optimization (and Proximal Gradient Descent). How much can clients (even benign ones) be trusted to perform a regularized training?

Minor:
a. The authors should consider using some of the standard notation in FL papers. This will simplify the presentation and improve the readability of the paper.
b. Line 168: W_{t, I}^{k}(\alpha=1) implies evaluating Eqn. (3) with \alpha=1. The authors mention that this shows that the k-th device is malicious. However, replacing \alpha=1 in Eqn. (3) implies that the agent is optimizing for the benign objective only. So, why is it considered malicious?

Some typos:
line 104: Often --> often
line 312: misclassification --> misclassification


**Time Spent Reviewing:**

7

---

> ### Author Response · Authors · 2021-08-10
> **Response to Reviewer ZHqh**
>
> Thank you for your valuable comments.
>
> Q1: Prior papers in this area including [8, 10, 11, 12] observe that the strength of an attack (even without a defense) starts to attenuate if there is no reinforcement by attackers in subsequent rounds. The benign updates are able to overwrite the effect of the attack. This is both intuitive and has been experimentally validated in the past. I really like the AEP analysis performed by the authors but I am not fully convinced, especially as it does not seem to reconcile with prior observations.
>
> A1: The observation of prior works is correct in most cases --  the strength of an attack starts to attenuate if there is no subsequent attacks launched by malicious clients  even without a defense. However, when the attack is extremely effective, this attenuation will be very slow and there is no guarantee that how many rounds are needed for the benign updates to overwrite the attack effect. In practice, the attack may be performed in multiple communication rounds. If the attack effect cannot be efficiently and effectively eliminated before the next attack, the attack effect can be accumulated and then results in worse training performance. We appreciate that you like our AEP analysis, and actually our AEP analysis is consistent with prior observations. As the title of Sec 4.3 indicates, we unveil the "long-lasting" rather than "permanent" attack effect. If the attack is effective (
> i.e. $\hat\delta_\tau$ is mainly in the kernel of $H_{t,i}^k$ for $t>\tau$), the changes between $\hat\delta_t$ and $\hat\delta_\tau$ will be small and the attack effect will mostly remain in the model. However, when the attack is not effective (i.e. $\hat\delta_\tau$ does not reside in the kernel of $H_{t,i}^k$ for $t>\tau$), the changes between $\hat\delta_t$ and $\hat\delta_\tau$ will be significant and the attack effect can be attenuated quickly by benign updates.
>
> Q2: The evaluation does not consider an adaptive attacker that is aware of the client-based optimization performed to remove the effect on the parameter subspace (hiding the attack).
>
> A2: We actually described in line 233-235 that the configurations of our defense strategy are hard to be determined by the attacker. However, we think it is a good idea to have a specific section discussing why our defense is not vulnerable to adaptive attackers, and we want to clarify the key reasons here. Some defenses are vulnerable to adaptive attacks because their  defensive operations are determined even before the attack is conducted. For example, when applying *Norm thresholding of updates*[7], the attacker knows the exact details of the defensive operation, i.e., the local updates with larger significance will be cropped. As such, an adaptive attacker can hide the attack effect in gradients with smaller significance by regularizing malicious updates. However, in our defense, the attacker cannot know the detailed defensive operation even after conducting the attack for three reasons: (1) Our defense is on the client side, and the defense happens during the local training after attacks. Since our detailed defensive process is closely related to other benign clients' data, which is inaccessible to the attackers, the attackers cannot determine the detailed defense process. (2) Even though the attackers have access to benign clients' data (which is a super strong assumption and beyond our threat model), the attackers cannot predict which benign clients will be sampled by the server to participate in training in the next communication round. (3) In the most extreme case where attackers have access to benign clients' data and can predict which clients will be sampled in the next round (which is an unrealistic assumption), the attacker still cannot successfully bypass our defense. The reason is that the defense during the benign local training is mainly dominated by the random noise $\Upsilon$ in Equ (13), which is also unpredictable. Such unpredictability and randomness make our defense strategy robust to the attackers, such that no effective attack can be adapted. In addition, no matter what adaptation the attacker made, the AEP still needs model parameter subspace to reside in. Since our defensive goal is to zero the dimension of the subspace that long-lasting AEP could reside in, any attack adaptation is hard to be effective. We will add an additional section to include the above clarifications regarding the adaptive attacker in the final version.
>
> Q3: The authors need to explicitly mention the threat model which will then provide the parameters for attacks that can be mounted on the defense. Furthermore, the solution requires clients participating in federated learning to perform a specific form of optimization (and Proximal Gradient Descent). How much can clients (even benign ones) be trusted to perform a regularized training?
>
> A3: Thanks for your constructive suggestion. We will elaborate our threat model more clearly in Sec 4.1 in the final version, and we  clarify the key points here. (1) The malicious attackers have the same knowledge as the benign clients except that they share a malicious dataset $D_M$. (2) The central server is benign and trusted. (3) All the benign clients will follow our proposed training protocol with our defense, i.e., performing our proposed regularized training.
>
> Minor 1: The authors should consider using some of the standard notation in FL papers. This will simplify the presentation and improve the readability of the paper.
>
> Answer: As reviewer2 writes, most of the notations are fully aligned with the state-of-the-art FL papers. The tricky part should be the notations related to AEP analysis. We will add a notation table to improve readability of the work.
>
> Minor 2: Line 168: $W_{t, I}^{k}(\alpha=1)$ implies evaluating Eqn. (3) with $\alpha=1$. The authors mention that this shows that the $k$-th device is malicious. However, replacing $\alpha=1$ in Eqn. (3) implies that the agent is optimizing for the benign objective only. So, why is it considered malicious?
>
> Answer: We apologize this typo, when $\alpha=1$, the $k$-th device is actually benign. We have corrected this in our new version.

---

> > ### Comment · Reviewer_ZHqh · 2021-08-30
> > **Post-rebuttal comments**
> >
> > I would like to thank the authors for the detailed clarifications. I am happy about most of the responses. However, I am not fully convinced with your response to Q1, especially, as I have not seen any empirical evidence towards this end.

---

> > > ### Author Response · Authors · 2021-08-31
> > > **Empirical evidience is included in Appendix B**
> > >
> > > Dear reviewer ZHqh,
> > >
> > > Thanks for your reply. Actually, we have empirical results showing that our theoretical analysis about AEP reconciles with previous works, which are provided in Appendix B. Figure 5 in Appendix B shows that the misclassification loss would increase even without any defense after the first attack, which reconciles with previous works. At the same time, Table 1 shows that the attack in Figure 5 would retain longer in the global model when $\Phi_t$ is smaller, which reconciles with our theoretical analysis. We think one potential cause that you thought our AEP analysis does not reconcile with previous works is the high attack frequency in our experiment settings of Section 7. When the attack happens frequently, the global model would not have enough rounds to eliminate the attack effect before the next adversarial round, and the attack effect would be accumulated in the global model.

---

### Public Comment · ~Song_Li4 · 2022-01-20
**Interested in the influence of noise injection on delta_t+1**

Dear authors:
I am recently reading your paper and interested in your work. I am a little confused about the delta updating rule which is Equation 7 in the paper. I understand that you can make H*delta_t non zero by injecting noise. But how can you ensure that injecting noise can make delta_t+1 smaller than delta_t as the noise is random. Further, how can the delta_t+1 diminish to 0 in a quicker speed, is there any theoretical guarantee?

---

### Decision · Program_Chairs · 2021-09-27

**Decision:**

Accept (Poster)

**Comment:**

This is an interesting paper, and the reviewers find much to recommend it.
Moreover, the rebuttal and discussion post-review seems to have addressed many, if not all, the concerns of the reviewers.
The issue of the attenuation of the attack was discussed. Perhaps the portion of the appendix where this is empirically addressed can be highlighted in the paper.